# Influence without Confounding: Causal Discovery from Temporal Data with Long-term Carry-over Effects

**Fan Li**[1,5,†]   **Zixuan Liu**[1,5,†]   **Yi Zhao**[2,*]   **Qi Tan**[3]   **Jinyang Li**[4]   **Ke Xu**[1,5,*]
**Shuihai Hu**[4]   **Jingbin Zhou**[4]   **Kun Tan**[4]
[1]Tsinghua University, Beijing, China   [2]Beijing Institute of Technology, Beijing, China
[3]Shenzhen University, Shenzhen, China   [4]Huawei Technologies Co., Ltd., China
[5]Zhongguancun Laboratory, Beijing, China

## Abstract

Learning causal structures from temporal data is fundamental to many practical tasks, such as physical laws discovery and root causes localization. Real-world systems often exhibit long-term carry-over effects, where the value of a variable at the current time can be influenced by distant past values of other variables. These effects, due to their large temporal span, are challenging to observe or model. Existing methods typically consider finite lag orders, which may lead to confounding from early historical data. Moreover, incorporating historical information often results in computational scalability issues. In this paper, we establish a theoretical framework for causal discovery in complex temporal scenarios where observational data exhibit long-term carry-over effects, and propose LEVER, a theoretically guaranteed novel causal discovery method for incomplete temporal data. Specifically, based on the *Limited-history Causal Identifiability Theorem*, we refine the variable values at each time step with data at a few preceding steps to mitigate long-term historical influences. Furthermore, we establish a theoretical connection between QR decomposition and causal discovery, and design an efficient reinforcement learning process to determine the optimal variable ordering. Finally, we recover the causal structure from the R matrix. We evaluate LEVER on both synthetic and real-world datasets. In static cases, LEVER reduces SHD by 17.29%-40.00% and improves the F1-score by 5.30%-8.79% compared to the best baseline. In temporal cases, it achieves a 64% reduction in SHD and a 45% improvement in F1-score. Additionally, LEVER demonstrates significantly higher precision on real-world data compared to baseline methods.

## 1 Introduction

Temporal causal discovery aims to recover the causal structure from time-series data. By revealing mechanistic rather than merely correlational dependencies, it provides a principled basis for reliable prediction (Kuang et al., 2018; Shen et al., 2018; Kuang et al., 2020; Peters et al., 2016), counterfactual reasoning (Swaminathan & Joachims, 2015; Wu & Wang, 2018; London & Sandler, 2019), and root-cause identification (Meng et al., 2020; Ikram et al., 2022; Lin et al., 2024).

A major challenge to temporal causal discovery is disentangling instantaneous influences from lagged ones across successive time steps. Many real-world time series exhibit lagged influences that persist far beyond a few time steps, giving rise to long-term carry-over effects. From the perspective of historical dependency, the value of a variable at a certain time depends not only on the current values of its parent variables, but also on their historical traces. Such systems are ubiquitous in both industrial production and daily life. For example, when an advertisement is launched, it stimulates consumers' purchase intentions, thereby driving up product sales in the following period. In this system, the current product sales are not determined solely by the advertising intensity at a single moment, but rather by the accumulated effect of advertising over a past period of time.

---

[†]Equal contribution   [*]Corresponding authors

Our goal is to recover the causal structure, i.e., a directed acyclic graph (DAG) that encodes the cause-and-effect relationships, from observational time-series data. Due to the prolonged duration of the carry-over effects, the influence of early historical data is often difficult to capture with limited observational data or model. Existing methods typically consider finite lag orders (Runge et al., 2019; Runge, 2020; Castri et al., 2023; Debeire et al., 2024; Saggioro et al., 2020; Sun et al., 2021a; Pamfil et al., 2020; Granger, 1969).

However, neglecting long-term carry-over effects may lead to historical values confounding causal discovery. Figure 1 illustrates a scenario of causal discovery using a model with a maximum lag order of 2, where observations of variables at different time steps are represented as temporal nodes. The historical node $X_2[t-3]$ simultaneously influences $X_1[t]$ and $X_3[t]$; however, as it is not included in the model, a spurious causal relationship may be inferred between $X_1[t]$ and $X_3[t]$ (Powell, 2018). Apart from the long-term carry-over effects, the super-exponential number of valid causal structures, reaching $4.2 \times 10^{18}$ for a graph with 10 nodes, poses an inherent challenge for causal discovery methods. Prior approaches that temporally unfold variables with a fixed lag order (Runge et al., 2019; Runge, 2020; Debeire et al., 2024) further exacerbate the search complexity.

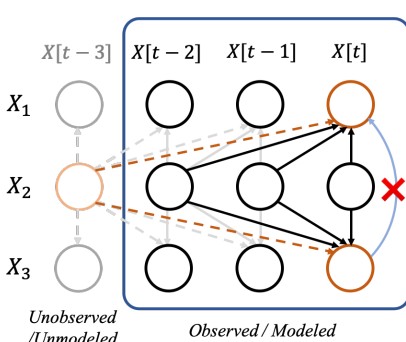

Figure 1: Historical confounding effect

In this paper, we establish a theoretical framework for causal discovery in temporal scenarios with long-term carry-over effects, and propose LEVER, a novel reinforcement learning (RL)-based causal discovery method grounded in this foundation. Specifically, we introduce a scoring function that guarantees causal identifiability in static settings. To address historical confounding in temporal data, we further establish the *Limited-History Causal Identifiability Theorem*, showing that refining observations with limited history suffices for maintaining causal identifiability. Guided by the above theory, we propose LEVER, which can discover the causal skeleton from incomplete observational temporal data. First, LEVER refines observations using a limited history to mitigate long-term confounding effects. It then employs an RL framework to determine the optimal variable ordering, using the upper-triangular R matrix from a QR decomposition as a compact and efficient state representation. Finally, the causal structure is recovered directly from the optimal R matrix.

We evaluate LEVER on both synthetic and real-world datasets. In static cases, LEVER reduces structural Hamming distance (SHD) by 17.29%-40.00% and improves the F1-score by 5.30%-8.79% compared to the best baseline. It achieves superior performance while requiring significantly less runtime and memory compared to baseline methods using the RL framework, demonstrating the effectiveness of using the R matrix as the state. In temporal cases, LEVER achieves a 64% reduction in SHD and a 45% improvement in F1-score. Compared to baseline methods for handling historical information, LEVER's historical refinement approach effectively addresses long-term carry-over effects without incurring significant additional overhead. Additionally, LEVER demonstrates significantly higher precision on real-world data compared to baselines.

Our contributions are summarized below:

- We extend static causal identifiability to temporal scenarios with long-term carry-over effects, and establish a theoretical connection between causal discovery and QR decomposition.

- We propose a causal discovery method for incomplete temporal data, which can quickly and effectively recover causal structure in the presence of long-term carry-over effects.

- We validate the effectiveness of our method on both synthetic and real-world dataset, demonstrating that our method outperforms the baselines in both accuracy and resource efficiency.

## 2 RELATED WORK

The PC algorithm (Spirtes et al., 2000) uses conditional independence tests to construct causal graphs but struggles with high-dimensional data due to combinatorial complexity. GES (Chickering,

2002) employs score-based optimization for efficient DAG learning. NOTEARS (Zheng et al., 2018) reformulates causal discovery as continuous optimization with an acyclicity constraint, enhancing scalability. SCORE (Rolland et al., 2022a) leverages the properties of the scoring function to achieve linear-time causal structure recovery, but this efficiency comes at the cost of shifting the computational burden to estimating the scoring function itself. RL-based approaches formulate causal discovery as a Markov decision process and use reinforcement learning to explore the combinatorial space of DAGs. RL (Zhu et al., 2019) generates a full adjacency matrix at each decision step, which can lead to high computational cost and convergence challenges. CORL (Wang et al., 2021) reduces the action space by reformulating static causal discovery as finding an optimal topological ordering, but its neural-network–based state encoder may introduce substantial computational overhead.

For temporal cases, although existing works on causal discovery have offered rigorous temporal modeling (Stein et al., 2025; Mastakouri et al., 2021), they typically assume that direct causal dependencies span only a finite duration. Granger causality (Granger, 1969) tests whether the past values of one variable help improve the prediction of another variable with given maximum time lag. PCMCI (Runge et al., 2019) and its variants (Runge, 2020; Debeire et al., 2024; Saggioro et al., 2020; Castri et al., 2023) extend PC to temporal settings by unfolding variables over fixed lags and employ momentary conditional independence tests to identify lagged causal relationships, yet exacerbate combinatorial complexity. NTS-NOTEARS (Sun et al., 2021a) and DYNOTEARS (Pamfil et al., 2020) introduce structural vector autoregressive model assumption with given model order, and solve an optimization problem with an acyclicity constraint. SyPI (Mastakouri et al., 2021) allows multi-lag dependency from historical data and provides a constrained-based causal feature selection method to determine the causes of an observed target variable. While these methods can effectively capture short-term lagged relationships, they do not consider the potential confounding effects arising from long-term carry-over influences.

There are also several works that address causal structure under subsampling or continuous-time kernels. Plis et al. (2015) and Abavisani et al. (2022) aim to learn the causal structure of a system with direct causal dependencies at a timescale $\tau_S$, given the causal graph at a slower timescale $\tau_T$. Bellot et al. (2021) assume that causal interactions occur between differential time steps and seeks to learn the causal structure of a continuous-time stochastic process from discretely sampled observations.

Under the assumptions of most existing work, distant historical values may influence the present only through observed intermediate time points. After conditioning on these mediators, the remote past is conditionally independent of the present and cannot act as a confounder for contemporaneous relations. In this work, we drop this assumption and allow distant history to exert direct causal effects on present variables. Such latent long-range causes can induce spurious dependencies among observed variables and make causal structure recovery more challenging.

## 3 THEOREM GUARANTEE

### 3.1 NOTATION AND SYSTEM MODEL

Let $X[t] \in \mathbb{R}^d$ denote the vector of *observations* (i.e., the values of all $d$ variables) at discrete time step $t \in \mathbb{N}$. We assume $X[t]$ is generated by a time-invariant system with *temporal carry-over effects*,

$$X[t] \;=\; \sum_{\tau=0}^{t} X[t-\tau]\,g(\tau) \;+\; \epsilon(t), \tag{1}$$

where $g(\tau) \in \mathbb{R}^{d \times d}$ is the impulse response matrix at lag $\tau$, and $\epsilon(t) \in \mathbb{R}^d$ is *structural noise* with i.i.d. entries.

**Assumption 1** (Time-invariant skeleton). *For every $\tau \geq 0$, the binary support $\mathrm{supp}(g(\tau)) := \{(i,j) \mid g_{ij} \neq 0\}$ is identical. That is, all $g(\tau)$ share the same directed acyclic skeleton $\mathcal{G}$.*

**Assumption 2** (Stationary lag dependency). *Every non-zero entry of $g(\tau)$ depends only on the lag $\tau$, not the absolute time index $t$.*

Assumptions 1-2 embody temporal ordering and causal invariance, which are standard in many existing works (Gong et al., 2015; 2017; Malinsky & Spirtes, 2018; Liu et al., 2023).

**Objective.** Given the observed time series $X[T_1:T_1 + T]$, our goal is to recover the causal skeleton $\mathcal{G}$ (i.e., the summary causal graph (Peters et al., 2013; Gong et al., 2024; Liu et al., 2023)). In the

following, we first introduce a scoring function and demonstrate that minimizing this score recovers the true causal topological order in a static setting, thereafter establish the static identifiability of the causal skeleton. We then extend the theoretical framework to partially observed temporal data, thereby aligning the results with our ultimate objective.

## 3.2 SCORING TOPOLOGICAL ORDERS

To identify the true causal topological order from all possible candidates, an ideal scoring function should attain its optimum at the true order. Following this principle, we introduce a scoring function based on the ordinary least squares (OLS) error.

Formally, let $Y \in \mathbb{R}^{m \times d}$ consist of $m$ $(m > d)$ i.i.d. observations of $d$ random variables $(X_1, \ldots, X_d)$ that follow Equation 1 under static setting, where $g(\tau) = 0$ for all $\tau > 0$. Let $\Pi$ be the set of all possible permutations of the $d$ variables, and define $K^\pi$ as the complete DAG whose skeleton is a complete graph and topological order is given by $\pi \in \Pi$.

**Lemma 1.** *For a set of $d$ variables, there exists a bijection between the set of all possible topological orders and the set of complete DAGs.*

**Definition 1** (Scoring function). *The score of a topological order $\pi$ given observations $Y$ is defined as the total OLS error of fitting each variable $j$ with its former variables in $\pi$ (denoted as $\mathrm{pa}^\pi(i)$):*

$$\mathrm{Score}(\pi; Y) := \sum_{j=1}^{d} \left\| Y_{:,j} - Y_{:,\mathrm{pa}^\pi(j)} \, \widehat{\beta}_j(K^\pi) \right\|_2^2, \tag{2}$$

*where $\widehat{\beta}_j(K^\pi) \in \mathbb{R}^{|\mathrm{pa}^\pi(j)| \times 1}$ are the OLS coefficients.*

**Theorem 1.** *Assume that $Y$ has full column rank. Let $Y^\pi \in \mathbb{R}^{m \times d}$ be the data matrix whose columns are permuted according to the given topological order $\pi$. Let $Y^\pi = QR$ be a QR decomposition of $Y^\pi$ with $R \in \mathbb{R}^{d \times d}$ upper–triangular and $\forall i, R_{i,i} > 0$. Then the score of $\pi$ can be expressed as*

$$\mathrm{Score}(\pi; Y) = \sum_{i=1}^{d} R_{i,i}^2. \tag{3}$$

**Theorem 2** (Causal topological order identifiability). *Let $\pi^*$ be a topological order of the true causal skeleton. For sufficiently large sample size $m$, the expected score of the true topological order $\pi^*$ almost surely attains the minimum over all possible topological orders $\pi \in \Pi$, i.e.,*

$$\mathbb{E}\big[\mathrm{Score}(\pi^*; Y)\big] \overset{a.s.}{=} \min_{\pi \in \Pi} \mathbb{E}\big[\mathrm{Score}(\pi; Y)\big]. \tag{4}$$

## 3.3 CAUSAL SKELETON RECOVERY

The complete DAG derived from the true causal topological order may include superfluous edges. We will demonstrate that the true causal graph can be recovered from the topological order through weight estimation and pruning.

**Theorem 3.** *Let $R \in \mathbb{R}^{d \times d}$ be the QR decomposition result of $Y^\pi$, and the index selector $\circ k$ represent the selection of the first $k$ elements. Let $W \in \mathbb{R}^{d \times d}$ be the estimated weight matrix, where $W_{i,j}$ represents the estimated edge weight from variable $i$ to variable $j$ if there exists a directed edge from node $i$ to node $j$ in $K^\pi$; otherwise, $W_{i,j} = 0$. Then for each variable $j$, the estimated weights from its parents to itself can be computed from $R$ as:*

$$W_{\mathrm{pa}^\pi(j),j} = \widehat{\beta}_j(K^\pi) = R_{\circ(j-1),\circ(j-1)}^{-1} \cdot R_{\circ(j-1),j}, \qquad 1 \leq j \leq d. \tag{5}$$

**Theorem 4.** *There exists a threshold $\theta > 0$ such that, when all edges in $K^{\pi^*}$ with absolute estimated weights $|W_{i,j}| < \theta$ are removed, the resulting graph is identical to the true causal graph.*

## 3.4 HISTORICAL CONFOUNDING IN TEMPORAL DATA

The aforementioned theorems establish causal identifiability in static scenarios. However, due to the carry-over effects in temporal data, these theories may not hold for the original temporal data. To address this, we propose a refinement scheme that mitigates the historical confounding effects, thereby extending causal identification theory to temporal settings.

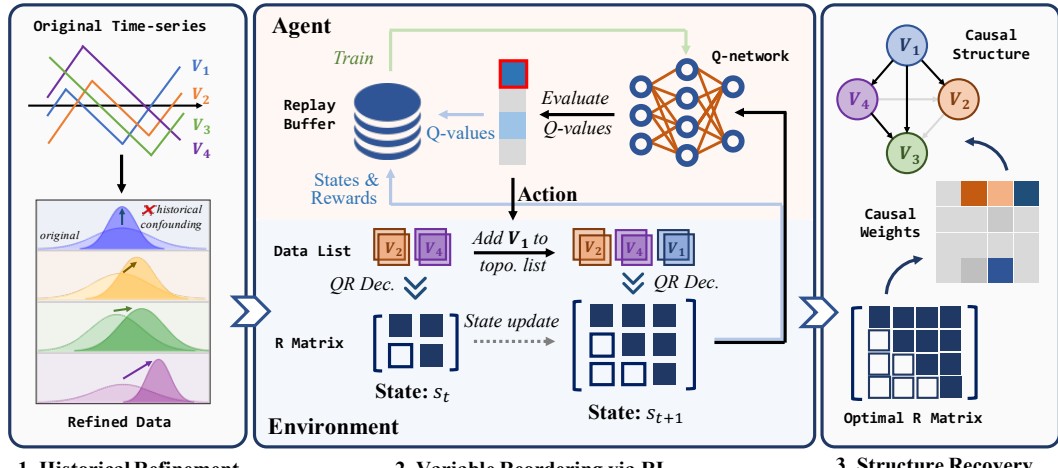

Figure 2: Overview of LEVER

**Theorem 5.** *Regressing $X[t]$ on its complete history $\{X[t-\tau]\}_{\tau \geq 1}$ and taking the residuals preserves causal identifiability, i.e., the true order minimizes score on these residuals.*

In the real world, one often has access to only incomplete history. We next show that, under a mild assumption, a limited history suffices.

**Assumption 3** (Linear recurrence of $g(\tau)$). *There exists $h \in \mathbb{N}$ and coefficients $k_1, \ldots, k_h \in \mathbb{R}$ such that $g(\tau) = k_1 g(\tau - 1) + \cdots + k_h g(\tau - h) \quad \forall \tau \geq h + 1$. A typical example is exponentially decaying effects, $g(\tau) = k\, g(\tau - 1)$.*

**Theorem 6** (**Limited-history Causal Identifiability**). *Let $\dot{Y}[t]$ be the residual obtained by regressing $X[t]$ on $X[t-1], \ldots, X[t-h]$. Under Assumption 3, the true topological order minimizes $\text{Score}(\pi; \dot{Y})$.*

## 4 LEVER

We propose LEVER, a method to learn the underlying causal structure from temporal data exhibiting long-term carry-over effects. Figure 2 illustrates the three key stages of LEVER. First, we apply a sliding window to extract samples from the temporal data and refine the values at the final time step of each window by regressing them on their preceding steps. Next, we utilize a Deep Q-Network (DQN) framework to efficiently determine the optimal variable ordering. We define the action at each decision step as appending the refined data of a specific variable to a data list. We perform QR decomposition on this data list to obtain the R matrix, which represents the state, and compute the change in global score as the reward. An episode concludes when all variables are included in the data list. Finally, we derive the causal weights from the optimal R matrix and prune edges with weights near zero to obtain the final causal structure.

### 4.1 HISTORICAL REFINEMENT

For observational data $X$ of length $T$, we use a sliding window of length $w$ to extract samples. Each sample comprises a historical observation matrix $H \in \mathbb{R}^{(w-1) \times d}$ and a target observation $\mathbf{x} \in \mathbb{R}^d$. By sliding the window with a step size $\Delta t$, we obtain $m = \lfloor \frac{T-w}{\Delta t} \rfloor + 1$ samples.

In practice, prior knowledge about the properties of the impulse response kernel $g(\tau)$ is typically unavailable. Thus, increasing the window size $w$ improves the likelihood of capturing a more complete structure of historical information. However, since the sample size $m$ decreases as $w$ increases, an excessively large $w$ may reduce the sample size, potentially compromising the accuracy of subsequent regression analysis and causal discovery. Therefore, a balance between historical information completeness and sample size should be considered to select an appropriate $w$.

We perform regression analysis using recent historical observations to refine the target observation. For each sample, we flatten its historical observation matrix into a $1 \times ((w-1) \cdot d)$ vector $H'$ by concatenating the values of the $d$ variables at each time step. We then regress the target observation $\mathbf{x}$ using the historical observation vector:

$$\arg \min_B \sum_{r=1}^{d} \|\mathbf{x}_r - H' B_r\|_2^2, \tag{6}$$

where $\mathbf{x}_r$ denotes the $r$-th variable of the target state, and $B_r \in \mathbb{R}^{(w-1) \cdot d \times 1}$ represents the $r$-th column of the regression coefficient matrix $B$.

Let the predicted value be $\hat{\mathbf{x}} = H'B$. The refined target state is then expressed as:

$$\dot{\mathbf{x}} = \mathbf{x} - \hat{\mathbf{x}} = \mathbf{x} - H'B. \tag{7}$$

Theorem 6 ensures that the true topological order achieves the minimal score on the residual $\dot{\mathbf{x}}$.

## 4.2 Variable Reordering via Reinforcement Learning

We employ a DQN framework to efficiently determine the optimal variable ordering.

**Action.** At each decision step, we define the action as appending the refined data of a specific variable to a data list. Since variables cannot be selected repeatedly, the action space at the $i$-th step is $\mathcal{A} := \{1, \cdots, d\} \setminus \Gamma$ with a size of $(d - i + 1)$, where $\Gamma$ is the indices of variables in current data list.

**State.** We define the state as the R matrix obtained from QR decomposition of the data list at the specific decision step. As established in the preceding theoretical analysis, the R matrix is uniquely determined by the data list and encapsulates complete information about the causal weights while being directly correlated with the score. Furthermore, QR decomposition can be performed stably and efficiently using numerical libraries such as NumPy (Van Der Walt et al., 2011) with a computational complexity of $O(md^2)$ (Golub & Van Loan, 2013). Thus, the R matrix serves as a compact and informative representation of the data list, accelerating the RL learning process and enhancing performance.

**Reward.** We compute the immediate reward of an action as the reduction in global score. The global score is defined as the sum of the residual sum of squares for variables within and outside the data list. Formally, it is expressed as:

$$\text{GlobalScore} = \sum_{i=1}^{|\Gamma|} R_{i,i}^2 + \sum_{j \notin \Gamma} \text{RSS}_{\Gamma \to j}, \tag{8}$$

where $R_{i,i}$ is the $i$-th diagonal element of the R matrix, and $\text{RSS}_{\Gamma \to j}$ denotes the residual sum of squares (RSS) when regressing $\dot{\mathbf{x}}_j$ on the data in the data list.

**Q-Network.** The Q-network maps the input environment state to the action with the highest expected return. To improve the accuracy of expected return estimation and ensure training stability, we employ a Double DQN (DDQN) approach (Van Hasselt et al., 2016). Specifically, one Q-network selects the optimal action, while a separate target Q-network evaluates the expected return for that action, reducing overestimation biases. Its parameters are updated using a temporal difference learning approach, with the update rule:

$$Q(s, a; \theta) \leftarrow Q(s, a; \theta) + \alpha \left( r + \gamma \max_{a'} Q(s', a'; \theta^-) - Q(s, a; \theta) \right), \tag{9}$$

where $s$ and $a$ denote the current state and action, $s'$ is the next state, $r$ is the reward, $\alpha$ is the learning rate, $\gamma$ is the discount factor, $\theta$ represents the Q-network parameters, and $\theta^-$ denotes the parameters of a target network. The target network's parameters $\theta^-$ are periodically aligned with $\theta$ using a soft update mechanism $\theta^- \leftarrow \lambda \theta + (1 - \lambda)\theta^-$, where $\lambda \in (0, 1)$ is a hyperparameter controlling the update rate.

During training, we track the episode with the highest cumulative reward and record the corresponding optimal R matrix $R^*$.

Table 1: Performance on static datasets

| Graph | Metrics | PC | NOTEARS | RL | CORL | SCORE | LEVER |
|-------|---------|-----|---------|-----|------|-------|-------|
| $G(10, 20)$ | F1 | 0.4574 | 0.7308 | 0.3809 | 0.5930 | **0.8179** | 0.8899 ▲8.79% |
| | FPR | 0.0949 | 0.0208 | 0.0580 | **0.0206** | 0.0329 | 0.0123 ▼40.96% |
| | SHD | 19.00 | 8.00 | 18.33 | 12.00 | **6.67** | 4.00 ▼40.00% |
| $G(20, 40)$ | F1 | 0.5528 | 0.7898 | 0.3770 | 0.5228 | **0.7918** | 0.8337 ▲5.30% |
| | FPR | 0.0395 | **0.0062** | 0.0193 | 0.0072 | 0.0207 | 0.0090 |
| | SHD | 32.00 | **13.54** | 34.00 | 25.85 | 15.50 | 11.60 ▼14.29% |

*Note: Black bold text indicates the best results among the baselines, and red triangle markers show the percentage improvement of our method compared to the best baseline results.*

## 4.3 STRUCTURE RECOVERY

We compute the causal weights $A$ using $R^*$ according to Equation 5:

$$W_{\mathrm{pa}^\pi(j),j} = R^{-1}_{\circ(j-1),\circ(j-1)} \cdot R_{\circ(j-1),j}, \qquad 1 \le j \le d.$$

We then prune edges with weights near zero to obtain a sparse causal structure. The impact of thresholding is illustrated in Figure 9.

## 5 EXPERIMENTS

**Implementation.** We evaluate LEVER on synthetic datasets, encompassing both static and temporal scenarios, as well as a real-world dataset, comparing its performance against eight classical and state-of-the-art baselines. All experiments are conducted on macOS with 16 GB of unified memory. The software environment utilizes Python 3.10.2 for scripting and PyTorch 2.5.1 as the primary deep learning framework. Implementation and configuration details are provided in the Appendix.

**Metrics.** We use Recall, False Positive Rate (FPR), F1-score and Structural Hamming distance (SHD) to evaluate the performance of causal discovery methods from multiple perspectives.

**Baselines.** The static causal discovery baselines include PC (Spirtes et al., 2000), NOTEARS (Zheng et al., 2018), RL (Zhu et al., 2019), CORL (Wang et al., 2021), and SCORE (Rolland et al., 2022a). The temporal causal discovery baselines include Granger (Granger, 1969), PCMCI (Runge et al., 2019), and NTS-NOTEARS (Sun et al., 2021a). We run baselines with multiple hyper-parameter settings, and report the results with best performance. The chosen hyper-parameters are presented in the Appendix.

### 5.1 SYNTHETIC STUDY

### 5.1.1 STATIC CASES

For static data, we generate the ground truth causal graphs with the Erdos-Renyi model. We denote the graphs with $d$ nodes and $m$ edges as $G(d, m)$. We generate 500 i.i.d. samples for each graph using the structural Equation $X_i = \sum_{j \in Pa(i)} X_j \cdot A_{ji} + \epsilon_i$, where $Pa(i)$ represents the topological parents of variable $i$ in the graph, $A_{ji}$ is the weight of the directed edge from variable $j$ to variable $i$, and $\epsilon_i$ is an independent random variable following a standard normal distribution. To ensure experimental reliability, we generate at least three datasets with different structures for each graph setting and report the average performance.

It's worth noting that although our primary focus is causal discovery on temporal data, static data can be considered a special case of temporal data, enabling comparison with a broader range of related work. Moreover, the performance of LEVER on static data validates the effectiveness of our RL module.

**Results.** Table 1 shows that LEVER achieves higher F1-scores and lower SHD compared to all baselines, demonstrating superior accuracy in causal structure recovery. Specifically, LEVER improves the F1-score by 5.30%-8.79% over the best-performing baseline and reduces SHD by 17.29%-40.00% relative to the lowest SHD among the baselines.

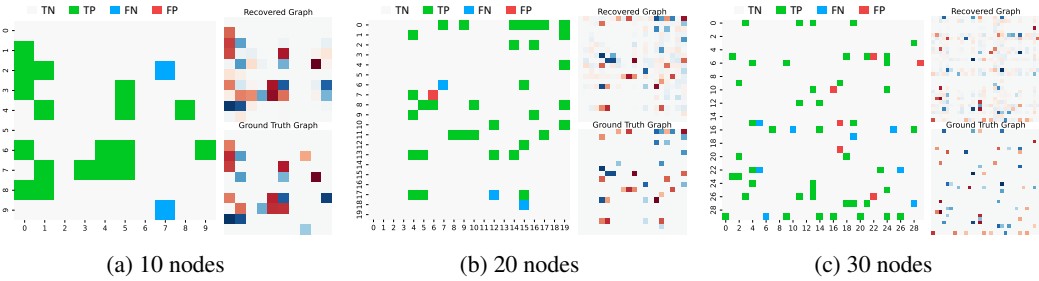

| (a) 10 nodes | (b) 20 nodes | (c) 30 nodes |

Figure 3: Visualization of LEVER's causal recovery

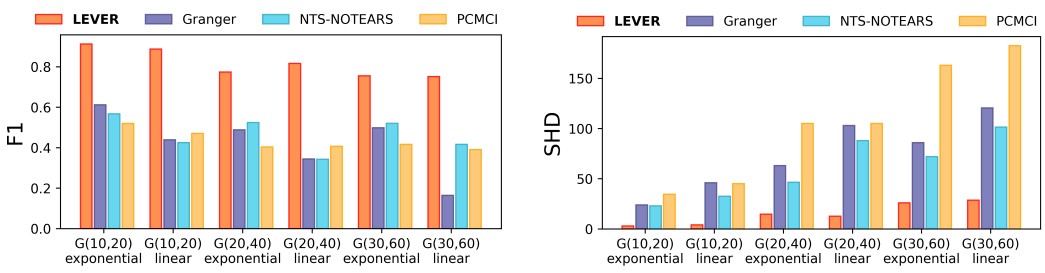

Figure 4: Performance on temporal datasets

Additionally, we compare the efficiency of LEVER with other RL-based causal discovery methods, namely RL and CORL. Owing to LEVER's action space of only $O(d)$ at each step and its efficient representation of raw data using the R matrix instead of a neural-network-based encoder, LEVER achieves superior efficiency in both time and memory usage. On the graph with 20 nodes, LEVER requires 0.25 seconds per episode, while RL and CORL require 19.45 seconds and 1.71 seconds, respectively. Furthermore, LEVER maintains a low peak memory usage of 10.36 MB, compared to 24.40 MB for RL and 64.68 MB for CORL (with input and hidden dimensions both set to 64).

### 5.1.2 TEMPORAL CASES

For temporal data with long-term carry-over effects, we generate the ground truth skeleton and instantaneous influence weights with the same settings as static dataset, and have the influence weights decay over lag time. We consider three types of decaying schemes, exponential, linear and complicated (specified later). We generate the original data with equation 1 for 2000 time steps, and use the data of last 1000 time steps, during which the system has been operating stably. At least three datasets are generated for each setting, and we report the average performance.

**Results.** Figure 3 visualizes the results of LEVER recovering causal relationships from temporal data with carry-over effects. It shows that LEVER can reconstruct the structure and weights of the ground truth causal graph with high accuracy. Figure 4 compares LEVER with baselines. LEVER achieves a 64%-88% reduction in SHD and a 45%-101% improvement in F1-score compared with the best baseline. Notably, NTS-NOTEARS and PCMCI, temporal extensions of NOTEARS and PC, incorporate limited lag-order temporal effects by unfolding variables across multiple time steps. This increases the number of nodes in the causal graph, significantly raising the time complexity. Under 10-node graph settings, the runtime of NOTEARS and PC on temporal data increases by 203 and 69 times compared to static data, respectively. In contrast, LEVER, which refines observations using historical data, incurs only an additional 0.0017 seconds of runtime.

### 5.2 DEEP DIVE

**Visualizing Long-term Causal Dependency** We generate two groups of datasets with complicated decay types, where $g(\tau)$ is linearly expressed by 10 and 20 preceding entries (i.e., $h = 10, 20$). Figure 5 presents heatmaps of $g(\tau)$ with different time lag $\tau = 10, 20, 50$. It can be observed that the actual carry-over effects persist over a much longer duration than the recurrence order $h$. Such long-term direct causal dependencies cannot be fully captured by a fixed maximum time-lag model

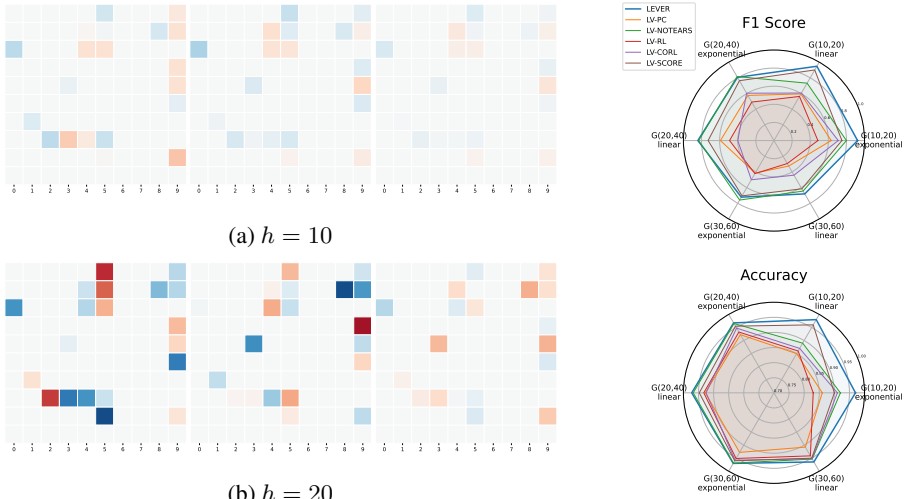

(a) $h = 10$

(b) $h = 20$

Figure 5: Heatmaps of $g(\tau)$ with $\tau = 10/20/50$

Figure 6: RL Effectiveness

as assumed in existing works, which may introduce confounding in causal discovery. In contrast, our method uses the $h$-step historical data as a *lever* to mitigate long-range historical confounding effects.

**Ablation Study.** We conduct an ablation study to assess the contributions of the RL-based variable reordering module in LEVER. We replace the RL-based variable reordering with static causal discovery baselines, and evaluate the performance of the modified LEVER on temporal datasets with both linear and exponential decay effects. Specifically, we consider five variants: LV-PC, LV-NOTEARS, LV-RL, LV-CORL, and LV-SCORE. Figure 6 compares the performance of these variants with the full LEVER model. The ablation results show that, although our RL ordering module is theoretically replaceable by existing algorithms in terms of objective, it achieves higher accuracy than these existing approaches.

**Sensitivity to Assumption Violations.** We evaluate LEVER on several modified temporal datasets where the values of three randomly selected causal dependencies are set to zero, except at lags of 3/5/10/25. The results in Table 2 show that the accuracy of our method does not decrease significantly even when the data violate the assumption. That is because the assumptions provide the necessary conditions for the theory to hold, yet certain scenarios outside these assumptions may also support the theory. Furthermore, operations such as pruning enhance our method's robustness in scenarios that deviate from the stated assumptions. These results support the generalization of our method to realistic cases.

Table 2: Results under assumption violation

| Dataset | SHD | F1 |
|---|---|---|
| Standard | 4 | 0.8947 |
| Modified (lag=3) | 5 | 0.8718 |
| Modified (lag=5) | 5 | 0.8718 |
| Modified (lag=10) | 3 | 0.9231 |
| Modified (lag=25) | 6 | 0.8421 |

## 5.3 REAL-WORLD STUDY

The *wt_walks_v1* dataset (Gamella et al., 2025b;a) contains temporal data derived from controlled experiments conducted in a wind-tunnel chamber. The ground truth graph comprises 16 nodes and 26 edges, representing the inherent dependencies among key variables within the wind-tunnel chamber. These variables include actuator settings (e.g., fan inputs and hatch position, denoted as $H$), sensor

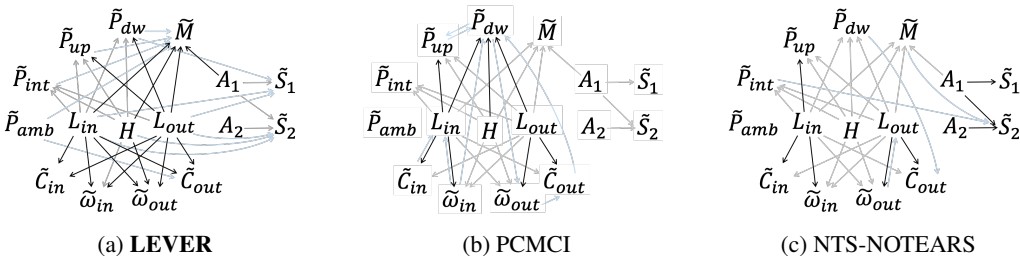

(a) **LEVER**          (b) PCMCI          (c) NTS-NOTEARS

Figure 7: Recovered causal structures on real-world datasets.

parameters (e.g., fan loads $L_{\text{in}}$, $L_{\text{out}}$), and sensor measurements (e.g., barometric pressures $\tilde{P}_{\text{dw}}$, $\tilde{P}_{\text{up}}$, $\tilde{P}_{\text{amb}}$, $\tilde{P}_{\text{int}}$).

We take the last 1000 observations for stability consideration. Figure 7 presents the ground truth dependency among variables and the recovered causal structures of LEVER and two baselines. Edges in black represent the recovered relations, edges in gray represent the undetected relations and edges in blue represent the relationships incorrectly inferred by the model. Quantitatively, LEVER achieves a SHD of 22 and an F1-score of 0.5769, outperforming PCMCI (SHD = 35, F1 = 0.3636), NTS-NOTEARS (SHD = 37, F1 = 0.3019), and Granger causality (SHD = 48, F1 = 0.2941). We do not visualize the result of the Granger causality method due to the excessive number of false positive edges it generates, which makes the causal graph overly cluttered and difficult to interpret.

## 6   LIMITATIONS

Our theory is built upon the linear time-invariant (LTI) assumption, which limits the theoretical guarantees of our approach in nonlinear scenarios. In future work, we will extend the LEVER framework to accommodate nonlinear relationships, thereby broadening its scope and applicability. Additionally, the assumption for using a few historical observations to completely eliminate long-term historical effects is that $g(\tau)$ can be linearly represented by $g(\tau - 1), \cdots, g(\tau - h)$. In future work, we will explore more relaxed conditions and quantify the impact of the long-term carry-over effects on causal discovery when these conditions are not fully satisfied.

## 7   CONCLUSION

In this work, we propose a causal discovery algorithm from temporal data with long-term carry-over effects. Our approach refines observations at each time step with data from a few preceding steps to mitigate long-term historical influences. We employ a reinforcement learning framework based on QR decomposition to determine the optimal variable ordering and the corresponding R matrix, from which we recover the causal structure in polynomial time. Experimental results show that our method outperforms baselines on both synthetic and real-world datasets.

## 8   REPRODUCIBILITY STATEMENT

We have made every effort to ensure that the results presented in this paper are reproducible. All code has been made publicly available at `https://github.com/Lifan1209/TemporalCausalRL.git` to facilitate replication and verification. The real-world dataset is publicly available in (Gamella et al., 2025b;a). The experimental setup, including training steps, model configurations, and hardware details, is described in detail in the paper (see Section 5 and Appendix A.2). We have also provided a full description of LEVER to assist others in reproducing our experiments. We believe these measures will enable other researchers to reproduce our work and further advance the field.

## 9 ACKNOWLEDGMENTS

This work was supported in part by the National Science Foundation for Distinguished Young Scholars of China (No. 62425201), the National Natural Science Foundation of China (No. 62472036), and Beijing Nova Program. Yi Zhao and Ke Xu are the corresponding authors.

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

## A   APPENDIX

### A.1   PROOFS FOR THEOREM GUARANTEE

#### A.1.1   PROOF OF LEMMA 1

*proof.* Let $V = \{1, \cdots, d\}$ be the variable set. Let $\Pi$ denote the set of all possible topological orders of $V$, and $\mathcal{K}$ denote the set of all complete DAGs on $V$.

Construct a map $f : \Pi \to \mathcal{K}$ as follows. For a topological order $\pi = (\pi_1, \cdots, \pi_d) \in \Pi$, let $f(\pi)$ be the directed graph on $V$ that has, for every pair $i < j$, the directed edge $\pi_i \to \pi_j$. Clearly every unordered pair of vertices receives exactly one oriented edge, so the underlying skeleton of $f(\pi)$ is

a complete graph. Because all edges point from earlier to later vertices in $\pi$, no directed cycle can exist; hence $f(\pi) \in \mathcal{K}$.

Define a mapping $g : \mathcal{K} \to \Pi$ such that, for any complete DAG $K \in \mathcal{K}$, $g(K)$ denotes the topological order of the nodes in $K$. Clearly $g(K) \in \Pi$. In addition, we can prove that any complete DAG $K$ has a unique topological order. Take any two distinct vertices $u, v$. In a complete DAG exactly one of the edges $u \to v$ or $v \to u$ is present. If $u \to v$, then in every topological order $u$ must precede $v$; otherwise $v$ must precede $u$. Thus the relative order of $u$ and $v$ is fixed. Since this holds for every unordered pair, the full order of $V$ is uniquely determined.

Finally, we verify that $f$ and $g$ are mutual inverses:

- For any $\pi \in \Pi$, the graph $f(\pi)$ was constructed by orienting each pair from earlier to later in $\pi$. The unique topological order of this graph is $\pi$ itself, hence $g(f(\pi)) = \pi$.

- For any $K \in \mathcal{K}$, its unique topological order $\pi = g(K)$ dictates that for every pair $\{u, v\}$ the orientation in $K$ matches the orientation in $f(\pi)$. Hence $f(g(K)) = K$.

Therefore, $f$ and $g$ are inverse bijections, completing the proof.

$\square$

### A.1.2 PROOF OF THEOREM 1

*proof.* Since performing QR decomposition on the reordered data matrix $Y^\pi$ is equivalent to applying Gram-Schmidt orthogonalization with order $\pi = (\pi_1, \cdots, \pi_d)$, the $j$-th diagonal element of the R matrix represents the norm of the orthogonal projection of $Y^\pi_{:,j}$ onto the complement of $\text{span}(Y^\pi_{:,\circ(j-1)})$. Let $S = \text{span}(Y^\pi_{:,\circ(j-1)})$, and $S^\perp$ represent the complement of $S$. Then,

$$R^2_{j,j} = \text{proj}_{S^\perp}(Y^\pi_{:,j}) = \left\| Y^\pi_{:,j} - Y^\pi_{:,\circ(j-1)} \widehat{\beta}_{\pi_j}(K^\pi) \right\|^2_2 = \left\| Y_{:,\pi_j} - Y_{:,\text{pa}^\pi(\pi_j)} \widehat{\beta}_{\pi_j}(K^\pi) \right\|^2_2,$$

Therefore, by definition, we have

$$\text{Score}(\pi; Y) = \sum_{j=1}^{d} R^2_{j,j}.$$

$\square$

### A.1.3 PROOF OF THEOREM 2

*proof.* Let $B \in \mathbb{R}^{d \times d}$ be the true weight matrix, where $B_{i,j}$ represents the true edge weight from variable $i$ to variable $j$ if there exists a directed edge from node $i$ to node $j$ in $K^\pi$; otherwise, $B_{i,j} = 0$. By definition, we have $Y = Y \cdot B + \epsilon$, where $\epsilon \in \mathbb{R}^{m \times d}$ is the structural noise with i.i.d. entries. Since $K^\pi$ is acyclic, $\det(I - B) = 1$.

Define $T = (I - B)^{-1}$, and $\Sigma = \sigma^2 T^\top T$. Then we have

$$\det\Sigma = \sigma^{2d} \cdot \det(T^\top) \cdot \det(T) = \sigma^{2d}.$$

Let $c_k(\pi) = \text{Var}(Y_{\pi_k} | Y_{\pi_1}, \cdots, Y_{\pi_{k-1}})$, and $\delta_k(\pi) = \frac{c_k(\pi)}{\sigma^2} > 0$. Let $\Sigma = LDL^\top$ be the LDL decomposition of $\Sigma$, then based on the properties of Schur complement, we have $D = \text{diag}(c_1(\pi), \cdots, c_d(\pi))$. Therefore,

$$\prod_{k=1}^{d} \delta_k(\pi) = \frac{\det\Sigma}{\sigma^{2d}}.$$

For correct order $\pi = \pi^*$, equations $c_k(\pi) = \sigma^2, \delta_k(\pi) = 1, \sum_{k=1}^{d} \delta_k(\pi) = d$ holds for every $k \in \{1, \cdots, d\}$. Clearly,

$$\prod_{k=1}^{d} \delta_k(\pi) = \frac{\det\Sigma}{\sigma^{2d}} = 1.$$

For incorrect order $\pi'$, since not all of $\{\delta_k(\pi')\}$ are equal to 1, it follows from the AM-GM inequality that $\sum_{k=1}^{d} \delta_k(\pi') > d$.

By definition, the score of topological order $\pi$ given $m > d$ independent observations $Y$ can be expressed as $Score_m(\pi; Y) = \sum_{k=1}^{d} RSS_k^{(\pi)}$. Then,

$$\frac{1}{m} Score_m(\pi; Y) = \sigma^2 \sum_{k=1}^{d} \frac{m-k+1}{m} \frac{RSS_k^{(\pi)}}{c_k(\pi)(m-k+1)} \delta_k(\pi).$$

According to the properties of Bartlett decomposition,

$$\frac{RSS_k^{(\pi)}}{c_k(\pi)(m-k+1)} = \frac{1}{m-k+1} \chi^2_{m-k+1},$$

where $\chi^2_{m-k+1}$ is a chi-squared random variable with $m-k+1$ degrees of freedom, i.e., the sum of squares of $m-k+1$ independent standard normal variables.

For each fixed $k$, the following equation holds for sufficiently large $m$:

$$\frac{1}{m-k+1} \chi^2_{m-k+1} = \frac{1}{m-k+1} \sum_{i=1}^{m-k+1} Z_{k,i}^2 \xrightarrow{a.s.} \mathbb{E}[Z_{k,1}^2] = 1,$$

where $Z_{k,i} \xrightarrow{i.i.d.} \mathcal{N}(0,1)$.

Furthermore, since $\frac{m-k+1}{m} \to 1$, we have

$$\frac{1}{m} Score_m(\pi; Y) \xrightarrow[m \to \infty]{a.s.} \sigma^2 \sum_{k=1}^{d} \delta_k(\pi).$$

Therefore, for correct order $\pi^*$, $\frac{1}{m} Score_m(\pi^*; Y) \xrightarrow[m \to \infty]{a.s.} \sigma^2 d$. For any incorrect order $\pi'$, $\frac{1}{m} Score_m(\pi'; Y) \xrightarrow[m \to \infty]{a.s.} \sigma^2 \sum_{k=1}^{d} \delta_k(\pi') > \sigma^2 d$.

As a result, the differential $\Delta_m = Score_m(\pi'; Y) - Score_m(\pi; Y)$ satisfies:

$$\frac{\Delta_m}{m} \xrightarrow[m \to \infty]{a.s.} c(\pi') := \sigma^2 \left( \sum_{k=1}^{d} \delta_k(\pi') - d \right) > 0.$$

Thus,

$$\Delta_m = c(\pi')m + o(m) \quad a.s.$$

$\square$

### A.1.4 PROOF OF THEOREM 3

*proof.* Since $\widehat{\beta}_j(K^\pi)$ are the OLS coefficients of fitting $Y_{:,j}$ on $Y_{:,\mathrm{pa}^\pi(j)}$, it can be presented as

$$\widehat{\beta}_j(K^\pi) = (Y_{:,\mathrm{pa}^\pi(j)}^\top Y_{:,\mathrm{pa}^\pi(j)})^{-1} Y_{:,\mathrm{pa}^\pi(j)}^\top Y_{:,j}$$

Let $Y^\pi = Q \cdot R$, then $Y_{:,\pi_j} = Q \cdot R_{:,j}$. Therefore,

$$\begin{aligned}
W_{\mathrm{pa}^\pi(j),j} &= \widehat{\beta}_j(K^\pi) \\
&= ((Q \cdot R_{:,\circ(j-1)})^\top \cdot (Q \cdot R_{:,\circ(j-1)}))^{-1} \cdot (Q \cdot R_{:,\circ(j-1)})^\top \cdot (Q \cdot R_{:,j}) \\
&= (R_{:,\circ(j-1)}^\top Q^\top Q R_{:,\circ(j-1)})^{-1} R_{:,\circ(j-1)}^\top Q^\top Q R_{:,j}
\end{aligned}$$

Since columns in $Q$ are normalized and mutually orthogonal, we have $Q^\top Q = I$. In addition, $R$ is upper-triangular. Then,

$$
\begin{aligned}
W_{\mathrm{pa}^\pi(j),j} &= (R_{:,\circ(j-1)}^\top R_{:,\circ(j-1)})^{-1} R_{:,\circ(j-1)}^\top R_{:,j} \\
&= (R_{\circ(j-1),\circ(j-1)}^\top R_{\circ(j-1),\circ(j-1)})^{-1} R_{\circ(j-1),\circ(j-1)}^\top R_{\circ(j-1),j} \\
&= R_{\circ(j-1),\circ(j-1)}^{-1} (R_{\circ(j-1),\circ(j-1)}^\top)^{-1} R_{\circ(j-1),\circ(j-1)}^\top R_{\circ(j-1),j} \\
&= R_{\circ(j-1),\circ(j-1)}^{-1} R_{\circ(j-1),j}
\end{aligned}
$$

$\square$

### A.1.5 PROOF OF THEOREM 4

*proof.* Let $\pi^*$ be the topological order of the true causal graph. Let $\{\mathrm{pa}^{\pi^*}(j)\}$ be the parents set of variable $j$ in the complete DAG $K^{\pi^*}$ (consisting with the notation in Definition 1), and $\{\widetilde{\mathrm{pa}}^{\pi^*}(j)\}$ be the parents set of variable $j$ in the true graph. Then for all $j \in \{1, \cdots, d\}$, $\{\widetilde{\mathrm{pa}}^{\pi^*}(j)\} \subseteq \{\mathrm{pa}^{\pi^*}(j)\}$.

Denote $\beta_j(K^{\pi^*})$ as the real coefficients, then $\widehat{\beta}_j(K^{\pi^*})$ are unbiased estimators of $\beta_j(K^{\pi^*})$, i.e., $\mathbb{E}[\widehat{\beta}_j(K^{\pi^*})] = \beta_j(K^{\pi^*})$, $\widehat{\beta}_j(K^{\pi^*}) \xrightarrow{m\to\infty} \beta_j(K^{\pi^*})$. Therefore, for any positive real number $\delta$, there exists an arbitrarily small $w \in (0, \delta/2)$ and an $m_0$ such that, for all $m > m_0$,

$$
\Pr\left(\forall k, j, \ |\widehat{\beta}_j(K^{\pi^*})_k - \beta_j(K^{\pi^*})_k| < w\right) > c.
$$

Moreover, for any $c < 1$, such an $m_0$ exists.

Denote $\Delta > 0$ as the minimum absolute value over all non-zero values in $\beta_j(K^{\pi^*})$. Let $\delta = \Delta$, then when $\forall k, j, |\widehat{\beta}_j(K^{\pi^*}) - \beta_j(K^{\pi^*})| < w$ happens, all zero elements in $\beta_j(K^{\pi^*})_k$ ($k \in \Omega^0$) satisfy $|\widehat{\beta}_j(K^{\pi^*})_k| < w$, and all non-zero elements in $\beta_j(K^{\pi^*})_k$ ($k \in \Omega$) satisfy $|\widehat{\beta}_j(K^{\pi^*})_k| \in (|\beta_j(K^{\pi^*})_k| - w, |\beta_j(K^{\pi^*})_k| + w)$.

Since $w < \delta/2$, we have $\forall k_1 \in \Omega^0, k_2 \in \Omega, |\widehat{\beta}_j(K^{\pi^*})_{k_1}| < |\widehat{\beta}_j(K^{\pi^*})_{k_2}|$.

Thus, there exists a threshold $\theta > 0$ such that, when all edges in $K^{\pi^*}$ with absolute estimated weights smaller than $\theta$ are removed, the resulting graph is identical to the true causal graph.

$\square$

### A.1.6 PROOF OF THEOREM 5

*Proof.* By definition, $X[t]$ is generated by the structural equation:

$$
X[t] = \sum_{\tau=0}^{t} X[t-\tau]g(\tau) + \epsilon(t),
$$

where $g(\tau) \in \mathbb{R}^{d \times d}$ is the impulse response matrix at lag $\tau$, and $\epsilon(t) \in \mathbb{R}^d$ is the structural noise vector with i.i.d. entries, following distribution $\mathcal{N}(0, \sigma^2)$. The causal graph $G$ is defined as the support pattern of the matrix $g(\tau)$.

Rewriting the equation above yields:

$$
X[t] = \sum_{\tau=1}^{t} X[t-\tau]g(\tau) + X[t]g(0) + \epsilon(t),
$$

$$
X[t](I - g(0)) = [X[t-1], X[t-2], \ldots, X[0]] \begin{bmatrix} g(1) \\ g(2) \\ \vdots \\ g(t) \end{bmatrix} + \epsilon(t). \tag{10}
$$

Thus, we obtain:

$$X[t] = [X[t-1], X[t-2], \ldots, X[0]]\beta + \epsilon(t)(I - g(0))^{-1},$$

where $\beta$ collects the lagged coefficient matrices.

Since $\epsilon(t)(I - g(0))^{-1}$ is independent of the regressors $[X[t-1], \ldots, X[0]]$, the residual of regressing $X[t]$ onto its past values is:

$$r[t] = \epsilon(t)(I - g(0))^{-1},$$

which can be transformed to the standard expression of the structural equation, i.e.,

$$r[t] = r[t]g(0) + \epsilon(t)$$

Then Theorem 2 and Theorem 4 can guarantee its identifiability.

$\square$

### A.1.7 Proof of Theorem 6

*Proof.* Let $S = [X[t-h], X[t-h+1], \ldots, X[t-1]]$ be the $h$ most recent historical observations, and let $U = [X[0], X[1], \ldots, X[t-h-1]]$ be the earlier historical observations.

Define the dependency weight vectors:

$$\Phi_U = \begin{bmatrix} g(t) \\ g(t-1) \\ \vdots \\ g(h+1) \end{bmatrix}, \quad \Phi_S = \begin{bmatrix} g(h) \\ g(h-1) \\ \vdots \\ g(1) \end{bmatrix},$$

where $\Phi_U$ and $\Phi_S$ represent the dependency weights of $U$ and $S$, respectively.

By Equation 10, $X[t]$ is expressed as a linear combination of historical observations and structural noise:

$$X[t] = U\Phi_U(I - g(0))^{-1} + S\Phi_S(I - g(0))^{-1} + \epsilon(t)(I - g(0))^{-1}.$$

Define:

$$W = (I - g(0))^{-1},$$

so that:

$$X[t] = U\Phi_U W + S\Phi_S W + \epsilon(t)W.$$

If the earlier observations $U$ are unavailable or not modeled, regressing $X[t]$ only on $S$, the residual is:

$$\begin{aligned} r[t]^+ &= (I - P_S)X[t] \\ &= (I - P_S)(U\Phi_U W + S\Phi_S W + \epsilon(t)W) \\ &= (I - P_S)U\Phi_U W + (I - P_S)S\Phi_S W + (I - P_S)\epsilon(t)W, \end{aligned}$$

where $P_S = S(S^T S)^{-1}S^T$ is the orthogonal projection matrix, and $I$ is the identity matrix. Since:

$$(I - P_S)S = S - S(S^T S)^{-1}S^T S = 0,$$

the residual simplifies to:

$$r[t]^+ = (I - P_S)U\Phi_U W + (I - P_S)\epsilon(t)W. \tag{11}$$

Define:

$$\Gamma_i = \begin{bmatrix} g(i+t-h) \\ \vdots \\ g(i+1) \end{bmatrix}.$$

By Assumption 3, for all $\tau \geq h+1$,

$$g(\tau) = k_1 g(\tau-1) + \cdots + k_h g(\tau-h),$$

implying:

$$\Gamma_h = k_1 \Gamma_{h-1} + k_2 \Gamma_{h-2} + \cdots + k_h \Gamma_0,$$

and by definition, $\Phi_U = \Gamma_h$.

Since:

$$X[t] = \sum_{\tau=0}^{t} X[t-\tau]g(\tau) + \epsilon(t),$$

we have:

$$X[t] = \left( \sum_{\tau=1}^{t} X[t-\tau]g(\tau) + \epsilon(t) \right) W.$$

Given $(I - P_S)S = 0$, it follows that:

$$(I - P_S)[X[t-h], X[t-h+1], \ldots, X[t-1]] = 0,$$

yielding:

$$(I - P_S)X[t-h] = 0, \tag{12}$$
$$(I - P_S)X[t-h+1] = 0, \tag{13}$$

$$\vdots$$

$$(I - P_S)X[t-1] = 0. \tag{14}$$

Unfolding $X[t-h]$ in Equation 12:

$$(I - P_S) \left( \sum_{\tau=1}^{t} X[t-h-\tau]g(\tau) + \epsilon(t-h) \right) W = 0.$$

Since $U = [X[0], X[1], \ldots, X[t-h-1]]$, this becomes:

$$(I - P_S)(U\Gamma_0 + \epsilon(t-h))W = 0.$$

As $W$ is invertible, we obtain:

$$(I - P_S)(U\Gamma_0 + \epsilon(t-h)) = 0.$$

Similarly, unfolding $X[t-h+1]$ in Equation 13:

$$(I - P_S) (U\Gamma_1 + X[t-h]g(1) + \epsilon(t-h+1)) = 0.$$

Using Equation 12, $(I - P_S)X[t-h] = 0$, this simplifies to:

$$(I - P_S)(U\Gamma_1 + \epsilon(t-h+1)) = 0.$$

Generalizing for $i = 1, \ldots, h$:

$$(I - P_S)(U\Gamma_i + \epsilon(t-h+i)) = 0. \tag{15}$$

Substitute Equation 15 into Equation 11. Since $\Phi_U = \Gamma_h$ and $\Gamma_h = \sum_{q=1}^{h} k_q \Gamma_{h-q}$, we have:

$$(I - P_S)U\Phi_U W = (I - P_S)U\Gamma_h(I - g(0))^{-1} = (I - P_S)U \left( \sum_{q=1}^{h} k_q \Gamma_{h-q} \right) W.$$

Using Equation 15, $(I - P_S)U\Gamma_{h-q} = -(I - P_S)\epsilon(t-q)$, so:

$$(I - P_S)U\Phi_U W = -(I - P_S) \sum_{q=1}^{h} k_q \epsilon(t-q)W.$$

Thus, Equation 11 becomes:

$$r[t]^+ = -(I - P_S) \sum_{q=1}^{h} k_q \epsilon(t-q)W + (I - P_S)\epsilon(t)W.$$

Rewrite:

$$r[t]^+(I - g(0)) = -(I - P_S)\left(\sum_{q=1}^{h} k_q \epsilon(t-q) + \epsilon(t)\right).$$

Define:

$$e[t] = (I - P_S)\left(\sum_{q=1}^{h} k_q \epsilon(t-q) + \epsilon(t)\right).$$

Then:

$$r[t]^+ = r[t]^+ g(0) - e[t].$$

Compute the expected squared norm of $e[t]$:

$$\mathbb{E}[\|e[t]\|_2^2] = \mathbb{E}\left[\left(\sum_{q=1}^{h} k_q \epsilon(t-q) + \epsilon(t)\right)^T (I - P_S)^T (I - P_S)\left(\sum_{q=1}^{h} k_q \epsilon(t-q) + \epsilon(t)\right)\right].$$

Since $I - P_S$ is symmetric and idempotent, and $\epsilon(t), \ldots, \epsilon(t-h)$ are independent, we have:

$$\mathbb{E}[\|e[t]\|_2^2] = \mathbb{E}\left[\epsilon(t)^T (I - P_S)\epsilon(t)\right] + \sum_{q=1}^{h} k_q^2 \mathbb{E}\left[\epsilon(t-q)^T (I - P_S)\epsilon(t-q)\right]$$

$$= \left(1 + \sum_{q=1}^{h} k_q^2\right)(m - s)\sigma^2 I_d, \tag{16}$$

where $I_d \in \mathbb{R}^{d \times d}$ is the identity matrix, $m$ is the number of samples, and $s = hd$ is the number of columns in $S$.

Equation 16 indicates that all columns of $e[t]$ have the same variance. Thus, the proof method from Theorem 2 can be applied to show that the true causal graph achieves the minimal score on $r[t]^+$. $\square$

## A.2 EXPERIMENT DETAILS

### A.2.1 LEVER IMPLEMENTATION

**Q-Network Structure and Training Hyper-parameters.** The Q-Network model consists of a normalized multi-layer perceptron with two hidden layers and ReLU activations. The input is first processed by a Normalization layer, then passed through two linear layers (with ReLU activation between them) followed by dropout. The final output layer maps the hidden representation back to the original input dimension. The mask is applied to the output to ensure that variables already in the data list are not selected again.

The values and description of hyper-parameters used in LEVER implementation are presented in Table 3. Our full implementation is publicly available at https://anonymous.4open.science/r/submit_TemporalCausalRL-BA08.

In addition, we evaluate all values in the output weight matrix as potential thresholds and select the one that yields the minimum SHD. The selected thresholds are presented in Table 4.

### A.2.2 BASELINES IMPLEMENTATION

We implement PC, GES, NOTEARS, RL, and CORL using the *gcastle* gca library and PCMCI using the *tigramite* tig library. Granger causality tests are performed with the `grangercausalitytests` function from the *statsmodels* sta package. For SCORE and NTS-NOTEARS, we use the original code provided by the authors Rolland et al. (2022b); Sun et al. (2021b). We run baselines with multiple hyper-parameter settings, and report the results with best performance.

For the **PC** algorithm, we set the confidence level to $\alpha = 0.05$ and use the Fisher-Z test as the conditional independence test.

Table 3: Q-Network hyper-parameters

| Parameter Name | Value | Description |
|---|---|---|
| window | 5 | Size of the sliding window used for processing sequential data. |
| sliding_step | 1 | Step size for advancing the sliding window. |
| buffer_size | 10000 | Capacity of the replay buffer for storing experience tuples. |
| minimal_size | 200 | Minimum number of experiences in the replay buffer required to start updating the Q-Network. |
| batch_size | 64 | Number of experience samples drawn from the replay buffer for each Q-Network update. |
| lr | 0.002 | Learning rate for optimizing the Q-Network during training. |
| hidden_dim | 16 | Number of units in the hidden layers of the Q-Network. |
| gamma | 0.9 | Discount factor determining the weight of future rewards in the Q-Network. |
| target_update | 2 | Frequency at which the target network's weights are updated to match the online Q-Network's weights. |
| epsilon_start | 1 | Initial exploration rate for Boltzmann exploration strategy. |
| epsilon_end | 0.01 | Final exploration rate for Boltzmann exploration strategy. |

Table 4: LEVER selected thresholds

| node_ct | decay_type | threshold |
|---|---|---|
| 10 | exp | 0.1398 |
| 10 | linear | 0.1720 |
| 20 | exp | 0.1279 |
| 20 | linear | 0.4627 |
| 30 | exp | 0.1372 |
| 30 | linear | 0.1223 |

For **NOTEARS**, we set the regularization parameter $\lambda_1 = 0.1$ and the maximum number of iterations to 1000. For each graph, we evaluate all values in the output matrix as potential thresholds and select the one that minimizes the Structural Hamming Distance (SHD). The selected thresholds are summarized in Table 5.

Table 5: NOTEARS selected thresholds

| node_ct | version | threshold | node_ct | version | threshold |
|---|---|---|---|---|---|
| 10 | v1 | 0.0105 | 20 | v6 | 0.0668 |
| 10 | v2 | 0.2268 | 20 | v7 | 0.1565 |
| 10 | v3 | 0.0019 | 20 | v8 | 0.1196 |
| 20 | v1 | 0.0374 | 20 | v9 | 0.0370 |
| 20 | v2 | 0.0748 | 20 | v10 | 0.0370 |
| 20 | v3 | 0.0484 | 20 | v11 | 0.1265 |
| 20 | v4 | 0.1032 | 20 | v12 | 0.0703 |
| 20 | v5 | 0.0684 | | | |

For **RL**, we set the number of training epochs to $nb\_epoch = 500$, the encoder input dimension to $input\_dimension = 64$, the encoder hidden dimension to $hidden\_dim = 64$, the decoder hidden

dimension to $decoder\_hidden\_dim = 16$, and the number of attention heads in the self-attention transformer to $num\_heads = 16$.

For **CORL**, we configure the batch size as $batch\_size = 128$, the encoder input dimension as $input\_dim = 128$, the embedding dimension as $embed\_dim = 128$, and the maximum number of iterations as $iteration = 100$.

For **SCORE**, we select the values of the regularization terms $\eta_G$ and $\eta_H$, as well as the pruning threshold $cutoff$, for each graph by minimizing the SHD. The selected values are reported in Table 6.

Table 6: SCORE selected hyper-parameters

| node_ct | version | eta_G | eta_H | cutoff | node_ct | version | eta_G | eta_H | cutoff |
|---------|---------|-------|-------|--------|---------|---------|-------|-------|--------|
| 10 | v1 | 0.05 | 0.0005 | 0.05 | 20 | v6 | 0.001 | 0.1 | 0.01 |
| 10 | v2 | 0.001 | 0.001 | 0.01 | 20 | v7 | 0.001 | 0.0005 | 0.01 |
| 10 | v3 | 0.001 | 0.0001 | 0.01 | 20 | v8 | 0.01 | 0.1 | 0.01 |
| 20 | v1 | 0.001 | 0.1 | 0.01 | 20 | v9 | 0.005 | 0.1 | 0.01 |
| 20 | v2 | 0.005 | 0.002 | 0.01 | 20 | v10 | 0.005 | 0.1 | 0.01 |
| 20 | v3 | 0.001 | 0.001 | 0.01 | 20 | v11 | 0.001 | 0.001 | 0.01 |
| 20 | v4 | 0.001 | 0.1 | 0.01 | 20 | v12 | 0.01 | 0.001 | 0.01 |
| 20 | v5 | 0.001 | 0.002 | 0.01 | | | | | |

For **PCMCI**, we use Partial Correlation test, and set the maximum time lag to $\tau\_max = 4$ to ensure the window size is consistent with our method, and set the confidence level to $\alpha = 0.01$.

For **Granger**, we set the maximum time lag to $maxlag = 4$ for the same reason, and use a significance threshold of $p\_thresh = 0.01$.

For **NTS-NOTEARS**, we set the regularization coefficients to $\lambda_1 = 0.005$ and $\lambda_2 = 0.01$, and select the threshold that minimizes the SHD. The selected values are reported in Table 7.

Table 7: NTS-NOTEARS selected thresholds

| node_ct | decay_type | threshold |
|---------|------------|-----------|
| 10 | exp | 0.1940 |
| 10 | linear | 0.1668 |
| 20 | exp | 0.1902 |
| 20 | linear | 0.1315 |
| 30 | exp | 0.1826 |
| 30 | linear | 0.1362 |

### A.2.3 EXPLANATION OF PERFORMANCE UNDER STATIC SCENARIOS

In Table 1, we observe that although the considered problems of LEVER and baselines are essentially the same, LEVER outperforms these baselines in static causal discovery tasks. This may be because, although both our method and the baselines require more data to achieve better performance, in practice the performance of our method is less sensitive to sample size than that of the baselines. As a result, our approach performs better when the available data are limited. To verify this, we conducted additional experiments in the static setting using 100, 200, 300, and 400 samples (the results reported in the main paper use 500 samples). As shown in Figure 8, our method has a more pronounced advantage when the sample size is small.

### A.2.4 IMPACT OF WINDOW SIZE.

We further investigate the impact of window size on algorithm accuracy relative to the theoretical minimum window size $h + 1$, which ensures complete elimination of historical effects. We evaluate LEVER's Receiver Operating Characteristic (ROC) curves under various window sizes on the datasets with complicated decay types where $g(\tau)$ is linearly expressed by 10 and 20 preceding

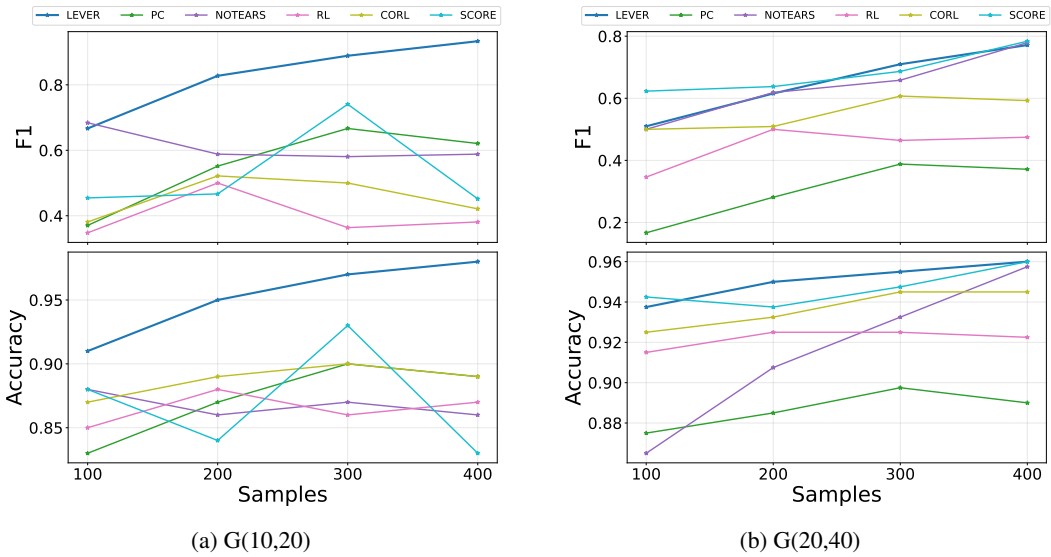

(a) G(10,20)          (b) G(20,40)

Figure 8: Performance comparison under different sample sizes in static scenarios.

entries. To isolate the effect of sample size, we use 500 samples for training under each window size configuration. As shown in Figure 9, when the window size is insufficient, the algorithm exhibits suboptimal performance. Increasing the window size effectively improves accuracy. However, when the window size exceeds $h + 1$, further increases yield less improvement in algorithm performance.

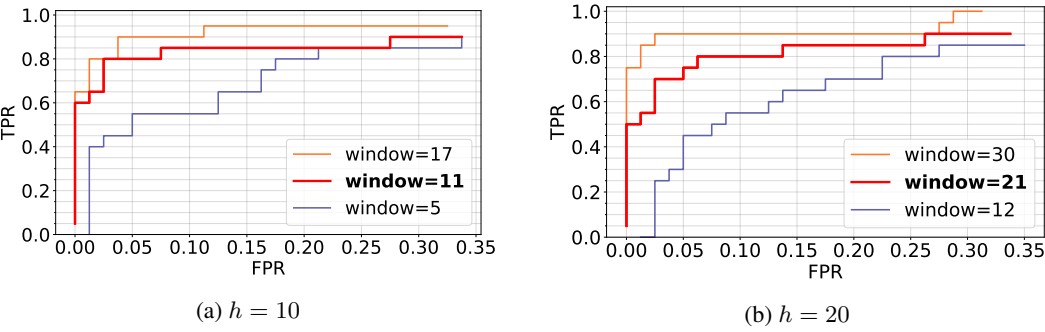

(a) $h = 10$          (b) $h = 20$

Figure 9: ROC curves

## B   LLM USAGE

Large Language Models (LLMs) were used to aid in the writing and polishing of the manuscript. Specifically, we used an LLM to assist in refining the language, improving readability, and ensuring clarity in various sections of the paper.

It is important to note that the LLM was not involved in the ideation, research methodology, or experimental design. All research concepts, ideas, and analyses were developed and conducted by the authors. The contributions of the LLM were solely focused on improving the linguistic quality of the paper, with no involvement in the scientific content or data analysis.

