# OpenReview forum: "Influence without Confounding: Causal Discovery from Temporal Data with Long-term Carry-over Effects"
_ICLR.cc/2026/Conference — ICLR 2026 Poster_

### Official Review · Reviewer_EqNz · 2025-11-01

**Soundness:** 3
**Presentation:** 3
**Contribution:** 3
**Rating:** 6
**Confidence:** 4

**Summary:**

This paper addresses the challenge of causal discovery in time series systems that exhibit long-term carry-over effects, where past influences persist far beyond a limited time lag and can create historical confounding. The authors show that instead of modeling the entire history, it is sufficient to regress out only a small recent window of past observations to obtain refined residual signals that preserve causal identifiability. Building on this result, they propose LEVER, a causal discovery algorithm that uses QR decomposition to represent variable relationships and employs reinforcement learning to efficiently search for the optimal causal ordering. Once the order is determined, causal edges are estimated and pruned to recover the final causal graph. Experiments on both synthetic and real-world datasets demonstrate that LEVER outperforms existing methods in accuracy and efficiency, particularly in scenarios with long-term temporal dependencies.

**Strengths:**

1. Addresses an important problem, long-term historical confounding, which is not adequately addressed in existing temporal causal discovery methods.
2. Instead of requiring full historical data (which is often unavailable, expensive, or noisy), the method works only with a small recent window, which is practically useful.
3. The method can work in both static and temporal settings.
4. The motivation and algorithmic pipeline are clearly explained.

**Weaknesses:**

1. The method has been evaluated only on one real-world dataset. Broader validation would strengthen claims of general applicability.
2. The method assumes the relationships between variables are linear and time-invariant. But in many real-world systems, relationships are often nonlinear.
3. The method requires choosing the window size, yet no automatic way to select it is given.

**Questions:**

1. How should practitioners choose the window size w in settings where the decay pattern of historical effects is unknown?
2. Can LEVER be extended to handle nonlinear causal relationships?
3. How does LEVER perform when some variables have very weak or delayed effects that only appear far beyond the chosen history window?
4. Is the learned causal ordering stable under subsampling?
5. Could LEVER be combined with domain knowledge, and how would that be incorporated into the RL reward or state representation?

---

> ### Author Response · Authors · 2025-11-23
>
> Thank you for the valuable feedbacks.
>
> ### **Performance with Long-term Weak or Delayed Effects:**
> Our method is specifically designed for scenarios with long-term carry-over effects. Rather than restricting causal dependencies to the selected history window, LEVER leverages information within the window to **remove the influence of distant historical confounding**, thereby enabling recovery of the underlying causal structure even when true effects occur beyond the observed horizon. The experimental results show that the recovered structure remains stable with long-term weak or delayed effects.
>
> ### **Real-world Evaluation:**
> We evaluate the effectiveness of our method against the baselines on the **wt_walks_v1** dataset. This real-world benchmark, published in _Nature Machine Intelligence_ (IF 18.8), provides a high level of reliability. It uses the physicistically validated dependencies among key variables in the wind-tunnel chamber as high-quality ground truth, and has been widely adopted in previous studies [1, 2, 3]. Therefore, the performance on the wt_walks_v1 dataset offers strong empirical evidence for the effectiveness of our approach.
>
> ### **Choosing Proper Window Size:**
> In settings where the decay pattern of historical effects is unknown, practitioners can select the window size $w$ using principled strategies:
> - **Stability analysis**, selecting the smallest $w$ beyond which the recovered structure remains unchanged;
> - **Cross-validation** when ground truth or downstream performance metrics are available.
> We plan to develop an **automatic window-selection procedure** in future work, potentially based on adaptive refinement or model-selection criteria.
>
> ### **Stability under Subsampling:**
> The subsampling problem—i.e., learning the causal structure of a system evolving at timescale $\tau_S$​ given measurements at a larger timescale $\tau_M$ [4]—is not explicitly addressed in the current framework. We intend to investigate the impact of subsampling and provide theoretical and empirical analyses in future work.
>
> ### **Combining with Domain Knowledge:**
> LEVER can incorporate domain knowledge in several ways. For example:
> - **Partial ordering constraints** can be enforced by adding penalty terms to the DQN objective;
> - **Forbidden or required edges** may be encoded through masking during action selection;
> - **Prior structural beliefs** can be injected into the state representation or reward shaping.
> These mechanisms can guide the learned ordering while preserving the flexibility of the reinforcement learning framework.
>
> ### **Extension to Nonlinear Cases:**
> Our theory is built upon the linear time-invariant (LTI) assumption, which limits the theoretical guarantees in nonlinear scenarios. In future work, we will extend the LEVER framework to accommodate nonlinear relationships.
>
> ### **References:**
> [1] Wehenkel A, Gamella J L, Sener O, et al. Addressing misspecification in simulation-based inference through data-driven calibration[J]. arXiv preprint arXiv:2405.08719, 2024.
> [2] Kern C, Fischer-Abaigar U, Schweisthal J, et al. Algorithms for reliable decision-making need causal reasoning[J]. Nature Computational Science, 2025, 5(5): 356-360.
> [3] Gu Y, Fang C, Bühlmann P, et al. Causality pursuit from heterogeneous environments via neural adversarial invariance learning[J]. The Annals of Statistics, 2025, 53(5): 2230-2257.
> [4] Plis S, Danks D, Freeman C, et al. Rate-agnostic (causal) structure learning[J]. Advances in neural information processing systems, 2015, 28.

---

> > ### Comment · Reviewer_EqNz · 2025-11-24
> >
> > I would like to thank the authors for addressing my comments and questions. After reviewing the author's rebuttal and other reviewers' comments, I will keep my initial score.

---

### Official Review · Reviewer_Kj1Y · 2025-11-01

**Soundness:** 2
**Presentation:** 3
**Contribution:** 3
**Rating:** 4
**Confidence:** 3

**Summary:**

The paper considers the problem of causal discovery from time-series data. Specifically, this authors address the case when the system may exhibit long-term carry-over effects, i.e., $X[t]$ may depend on $X[t-h]$ for large or even infinite $h$. Most existing methods assume that the lag order is finite; hence, applying these methods to the considered setting may lead to spurious edges in the recovered causal graph. The authors first show that, for static data (i.e., when only instantaneous causal effects exist), the correct causal ordering can be identified by performing QR decomposition on the observed data. Furthermore, for temporal data, they show that by assuming that the infinite effect function can be summarized by a linear combination of finite effects, the correct causal ordering can be recovered by first regressing $X[t]$ on limited-history data $(X[t-1],\cdots, X[t-h])$ and then applying the method for static data to the residuals. The authors develop an identification algorithm based on the theoretical results and evaluate its performance on both synthetic and real datasets.

**Strengths:**

1. Clear explanation of all theoretical results, along with motivating examples and detailed proofs.
2. The overall presentation is clear and easy to follow.
3. The authors conduct extensive simulations to demonstrate the performance of the proposed algorithm.

**Weaknesses:**

The main weakness is that the considered setting substantially overlaps with existing work on causal structure learning. In the case with static data, the problem reduces to causal discovery in linear SCMs with i.i.d. noise distributions. It has already been shown that the model is uniquely identifiable under either Gaussian or non-Gaussian noise distributions, and many existing algorithms can be applied to this setting for causal structure learning. In the case of temporal data, the authors show that, under Assumptions 1, 2 and 3, there exists an $h$ such that the residuals can be written in the form of a linear SCM with equal noise variances (line 901). Further, in the proposed algorithm, it seems that the RL component could also be replaced by existing algorithms (see Q4 below).

**Questions:**

1. Should Equation (1) be $g(\tau) X[t-\tau]$, since $g(\tau)$ is a $d\times d$ matrix?
2. In the case of static data, is there a reason why the proposed method outperforms existing approaches (e.g., NOTEARS), especially in the simulation results? My understanding is that the considered problems are essentially the same.
3.  In Section 4.1, does the optimal selection of $w$ always correspond to the $h$ in Assumption 3? If not, I am not sure whether the results in Theorem 6 can be directly applied to the residual $\dot{\mathbf{x}}$ in Equation (7).
4. Can the DQN framework (i.e., steps 2 and 3 in Figure 2) be replaced by existing causal structure learning algorithms?
5. Is the existence of time-dependent effects (i.e., whether the data are static or temporal) provided as an input under the different settings in the simulation results?

---

> ### Author Response · Authors · 2025-11-23
>
> Thank you for the valuable comments.
>
> ### **Highlighting Our Contribution:**
> The primary contribution of our work lies in addressing causal discovery in temporal settings with long-term carry-over effects. The discussion of the static scenario serves to establish causal identifiability under our proposed scoring function, which is ultimately applied for causal discovery in temporal scenarios.
>
>
> ### **Comparing with Existing Static Causal Discovery Methods:**
> Our causal discovery objective in the static setting is aligned with that of mainstream static causal discovery methods. However, experiments in the static scenario show that our approach outperforms existing methods. A possible reason is that, although both our method and the baselines benefit from larger sample sizes, in practice the performance of our approach is less sensitive to sample size. As a result, our method achieves better performance when the available data are limited.
>
> To verify this, we conducted additional experiments in the static setting using 100, 200, 300, and 400 samples. The experimental results show that our method has a more pronounced advantage when the sample size is small. We include these experiments and results in the appendix of the revision.
>
> ---
> Sample size,   Graph size,    Method,         F1,                  FPR,           SHD
>
> 100,                  G(20,40),      LEVER,          0.5098,          0.0055,          25
>
> 100,                  G(20,40),      NOTEARS,    0.5,                 0.1236,          54
>
> 200,                  G(20,40),      LEVER,          0.6154,          0.0,             20
>
> 200,                  G(20,40),      NOTEARS,    0.6186,          0.0852,          37
>
> 300,                  G(20,40),      LEVER,          0.7097,          0.011,           18
>
> 300,                  G(20,40),      NOTEARS,    0.6582,          0.0467,          27
>
> 400,                  G(20,40),      LEVER,          0.7714,          0.0192,          16
>
> 400,                  G(20,40),      NOTEARS,    0.7792,          0.0302,          17
>
> ---
>
> ### **Expression in Equation (1):**
> We treat $X[t]$ as a row vector; therefore, it is right-multiplied by $g(\tau)$.
>
> ### **The Optimal Selection of $w$:**
> When Assumption 3 holds exactly, $h+1$ is the optimal choice of $w$. However, in broader scenarios where Assumption 3 only approximately holds, using a larger $w$ can help improve accuracy to some extent (although this may come at the cost of reduced sample size).
>
> ### **Replaceability of the DQN Framework:**
> We have added an ablation study on the DQN framework in the revision. The results show that LEVER equipped with the DQN framework achieves higher accuracy than the modified versions in which the RL module is replaced by existing static causal discovery methods.
>
> ### **Whether Time-Dependent Effects Are Provided as Input in the Simulation Results**
> Yes, the existence of time-dependent effects is provided as input. It is reasonable to assume that practitioners know in prior whether the problem they are dealing with is temporal or static.

---

### Official Review · Reviewer_1PFq · 2025-11-03

**Soundness:** 2
**Presentation:** 2
**Contribution:** 2
**Rating:** 2
**Confidence:** 4

**Summary:**

This work introduce a method called "LEVER", which uses a (QR-decomposition)-based, "static" score and RL to infer a complete topological (causal) order; based on i.i.d. samples from time series. The theory part mainly aims at "static skeleton recovery", assuming i.i.d. samples, and then consider "historical refinement" via regression of variables on their past values. The baselines of the empirical experiment consider "static discovery methods" and embed them into the LEVER framework (of Figure 2), and a couple of temporal baselines (not a comprehensive list, missing main recent ones).

Overall: The only "temporal" modeling of the theory part comes in the historical refinement framework, unlike other recent studies, e.g., [Mastakouri, Schölkopf, Janzing ICML 21]; [Sun,Schulte,Guiliang,Pascal,Poupart AISTATs 23]; [Liu, Sun, Hu, Wang, NeurIPS23]; and [Stein,Shadaydeh,Blunk,Penzel,Denzler ICLR 25]. Also, temporal baselines are limited. I see the contribution is more limited to static skeleton discovery, unlike what the current presentation of the paper suggests. Please see more details in the Weaknesses section.

**Strengths:**

The QR-based score function for static skelton learning, as formalized and used in LEVER, and supported by theory in Sections 3.1-3.3, is novel, up to the best of my knowledge.

**Weaknesses:**

===The theory part===

- Abstract & intro give the impression that the proposed method solves for a "full" causal discovery problem, including recovering directions; but objective, theory, and experiments sections mainly solve for "causal skeleton (adjacency matrix)  recovery"; while rely on limited connections between the two.  Lemma 1 discusses a bijection between the sets of "exact" topological orders & "complete" DAGs, i.e., those with all edges directed consistently with a given full ordering. This follows from classic results in graph theory but it does not certainly address the complexity of the "full" causal discovery problem, where DAGs could be still recovered based on partial topological orders. Theorems 3&4 determine edge coefficients based on the same assumptions.

- Sec 3.1-3.3 do not seem to model for time-series data including the proposed score. Sections 3.2 & 3.3 explicitly assume i.i.d. samples as stated in the 2nd para of Sec 3.2 ``Formally, let $Y \in R^{m\times  d}$ consists of $m (m > d)$ $\textbf{i.i.d.}$ observations of $d$ random variables''. This assumption is clear in Thm 1 for "Full column rank" of data matrix $Y$, the proposed score in Eq (3), and Thms 3&4.  While such an assumption is common for "static" causal discovery, it shouldn't be assumed when modeling "temporal" dependence.  Sec 3.4 then introduce historical confounding by regressing $X[t]$ on its complete history $\{X[t−\tau]\}_{\tau \geq 1}$ and taking the residuals. This is the only place where temporal modeling is assumed.

- There is no explicit relation between "the theory proposed by authors" and "existing work on the same topic" is provided. Existing work, which provide more rigorous temporal modeling in their theory include: [Mastakouri, Schölkopf, Janzing ICML 21];[Sun,Schulte,Guiliang,Pascal,Poupart AISTATs 23]; [Liu, Sun, Hu, Wang, NeurIPS23]; [Stein,Shadaydeh,Blunk,Penzel,Denzler ICLR 2025]. Only [Sun,Schulte,Guiliang,Pascal,Poupart AISTATs2023] & [Liu, Sun, Hu, Wang, NeurIPS23] are cited in this submission, the other two are not.

=== Experiments === The only temporal baseline considered is [Sun,Schulte,Guiliang,Pascal,Poupart AISTATs2023].

**Questions:**

I don't think reference [Liu et al., NeurIPS 2023] assumes either "time-invariant skeleton" nor "stationary lag-dependency" as claimed by authors below Assumptions 1 & 2, no?

---

> ### Author Response · Authors · 2025-11-23
>
> Thank you for your thoughtful comments.
>
> ### **Clarification of the Problem Addressed in This Work:**
> We apologize for any confusion caused by our description. In fact, our work addresses the complete causal discovery problem, including the recovery of causal directions. In the final adjacency matrix $W$ produced by our algorithm, $W_{i,j} \neq 0$ indicates that there is a directed edge from node $i$ to node $j$ in the recovered summary graph, whereas $W_{i,j} = 0$ means that no such edge exists. This objective and graphical representation are consistent with mainstream approaches in temporal causal discovery (Peters et al., 2013; Gong et al., 2024; Liu et al., 2023). We have refined the expression of our goal in the revision.
>
> ### **Relationship Between the Static and Temporal Scenarios:**
> The ultimate goal of our work is to recover the summary graph from time-series data. To achieve this, we propose a scoring function and establish causal identifiability under this score in the static setting (Sections 3.2 and 3.3). In Section 3.4, we further show that the confounding effect of long-term history can be mitigated through a historical refinement scheme, which transforms the temporal causal discovery problem into static one. Consequently, we develop the LEVER algorithm, which applies the results from Sections 3.2 and 3.3 to the refined time-series data to address the temporal causal discovery problem.
>
>
> ### **Positioning Our Work Among Existing Research on the Same Topic:**
> In the Related Work section, we have included a range of temporal causal discovery methods, such as Granger causality (Granger, 1969), PCMCI (Runge et al., 2019) and its variants (Runge, 2020; Debeire et al., 2024; Saggioro et al., 2020; Castri et al., 2023), NTS-NOTEARS (Sun et al., 2021a), and DYNOTEARS (Pamfil et al., 2020). We also point out that existing temporal causal discovery approaches typically assume a finite maximum time lag. While these methods can effectively capture short-term lagged relationships, they do not consider the potential confounding effects arising from long-term carry-over influences, which is the problem our work aims to address. Thus, in contrast to these approaches, our temporal modeling framework additionally allows historical values at infinitely long time lags to directly affect current variable values.
>
> Thank you for providing additional references on temporal causal discovery. Among the works we did not cite, [Mastakouri, Schölkopf, Janzing, ICML 2021] also assumes a finite maximum time lag (as implied in Algorithm 1, line 4).  [Stein, Shadaydeh, Blunk, Penzel, Denzler, ICLR 2025] introduces a real-world causal discovery benchmarking kit for time-series data, but does not propose a new causal discovery method. We include these references in the related work section in the revision.
>
>
> ### **Temporal Baselines:**
> We have included the Granger causality test (Granger, 1969), PCMCI (Runge et al., 2019), and NTS-NOTEARS (Sun et al., 2021a) as temporal baselines (see the paragraph “Baselines” before Section 5.1).
>
> ### **“Time-invariant Skeleton” and “Stationary Lag-dependency” Assumptions in [Liu et al., NeurIPS 2023] :**
> [Liu et al., NeurIPS 2023] assumes both of these assumptions.  In Section 2, paragraph 3 of [Liu et al., NeurIPS 2023], the authors state that “Implicit in (1) is the assumption that cause precedes effect, and that causation is invariant to time,” which corresponds to our **stationary lag-dependency assumption**. Moreover, since [Liu et al., NeurIPS 2023] assumes that causal relations exist only between adjacent time steps, their assumption can be seen as a special case of our **time-invariant skeleton assumption**.

---

### Official Review · Reviewer_eY1p · 2025-11-11

**Soundness:** 4
**Presentation:** 3
**Contribution:** 4
**Rating:** 8
**Confidence:** 4

**Summary:**

The paper studies causal-structure recovery from time series with long-term carry-over effects. It (1) proves that a score based on OLS/QR identifies the true topological order in the static case, (2) shows that regressing out a limited recent history suffices under a linear-recurrence assumption to remove historical confounding, (3) uses the R matrix from QR as an RL-friendly compact state and trains a DQN to recover a topological ordering, and (4) recovers edges from the optimal R and prunes small weights. Theory, algorithm, and strong empirical gains on synthetic and a wind-tunnel dataset are presented.

**Strengths:**

• Clean, provable identifiability result linking QR diagonal terms and the order score. Theorems 1–4 give solid, interpretable mechanics for order scoring and weight recovery.

• Practical mechanism to mitigate long-range confounding via residualization on limited history and a clear sufficient condition (linear recurrence) under which limited history suffices. The Limited-history Causal Identifiability theorem is useful.

• Engineering innovation: using the R matrix as an RL state is simple, computationally efficient, and justified by the theory. This yields large runtime and memory savings versus prior RL encoders.

• Strong empirical performance. The method outperforms several static and temporal baselines in F1 and SHD and shows large improvements in temporal scenarios. Results are reported on multiple synthetic regimes and a real dataset.

• Reproducibility: code and dataset links and implementation details are noted in the appendix.

**Weaknesses:**

1. The related-work section treats prior temporal methods as finite-lag. But there is a body of work that models dependencies without imposing a strict finite lag (rate-agnostic or effectively infinite-memory representations). The paper should explicitly compare and position itself relative to those methods (both conceptually and empirically when possible). Examples the authors did not cite (add and discuss):
- Rate-Agnostic (Causal) Structure Learning — Sergey Plis et al. — 2015.
- Generalized Rate-Agnostic Causal Estimation via Constraints (GRACE-C) — M. Abavisani et al. 2023
- Consistency of Mechanistic Causal Discovery in Continuous Time (Neural ODEs) — A Bellot et al. 2022
, plus other works that address causal structure under subsampling or continuous-time kernels. Including these clarifies novelty and assumptions.

2. The Limited-history theorem depends on Assumption 3 (linear recurrence of g(τ)). The paper acknowledges this, but lacks an empirical sensitivity analysis showing how performance degrades when the linear-recurrence assumption is violated or only approximately holds. A stronger empirical section on robustness is needed.

3. Figure 3 is hard to parse and compare to ground truth. The visualization layout, legends, and the mapping between recovered and true edges are not easy to read.

4. The RL ordering module is novel here, but the paper would benefit from (a) an ablation that replaces RL ordering with a simple heuristic (e.g., greedy order by marginal variance or SCORE ordering) to quantify the RL contribution and (b) more baselines that explicitly handle long memory when available

**Questions:**

1. How sensitive is LEVER when Assumption 3 is only approximately true? Please include an experiment where g(τ) is not exactly linear-recurrent.

2. The RL state uses the R matrix. How does numerical stability affect performance when columns are nearly collinear? Do you need column normalization or regularization in practice?

---

> ### Author Response · Authors · 2025-11-23
>
> Thank you for your valuable suggestions.
>
> ### **Comparison with Existing Methods:**
> The problem we consider is substantially different from existing works that address causal structure under subsampling or continuous-time kernels.
> **First, our setting differs from these studies.** Prior works aim to learn the causal structure of a system that evolves at a timescale $τ_S$, given measurements at a slower timescale $τ_M$ [1,2,3], or aim to learn the causal structure of a continuous-time stochastic process from discretely sampled observations [4]. In contrast, we study a setting where the value of each variable is subject to lagged causal influences from long-term historical values, yet the available observational trace is not long enough to eliminate spurious associations induced by such delayed causal effects.
>
> **Moreover, our assumptions fundamentally differ from the above works**, which poses additional challenges in our case. **Existing studies typically assume that causal interactions occur only between adjacent (differential) time steps** [1,2,3,4]. Note that although under this assumption, long-term (≫ measurement timescale) historical values can influence the present, **the influence  is intermediated by variables at observable time steps**. This implies that the unsampled historical values are conditionally independent of the present given the variables at intermediate observation time steps. Consequently, conditioned on the observed data, the unsampled historical values do not act as confounders that distort causal structure estimation.  In contrast, we consider settings where historical values may have **direct** causal effects on the present. In this case, the unsampled historical variables may induce spurious causal relationships among observed variables (as described in our Introduction), significantly increasing the difficulty of the problem. Our main contribution is to establish identifiability for causal discovery in the presence of such historical confounding, and to develop an efficient temporal causal discovery method guided by this theory.
>
> In addition, the algorithms in [1,2] take a graph $H$ defined at the measurement timescale as input (as stated in [2] : “whether learned from data $D$, expert domain knowledge, both of these, or some other source”), without addressing how to correctly derive $H$ from temporal observational data—an issue that is both important and nontrivial. This is precisely where our work focuses.
>
> We have included the discussion above in the revision.
>
> ### **Sensitivity Analysis on the Approximately Valid Linear-Recurrence Assumption:**
> We have included experimental results under assumption violation in Appendix A.2.3. The results show that the accuracy of our method does not degrade significantly even when the data violate the assumption. This is because the assumptions provide the necessary conditions for the theory to hold, while certain scenarios outside these assumptions may still satisfy the theoretical requirements. Moreover, operations such as pruning further enhance the robustness of our method in settings that deviate from the stated assumptions. These results support the generalizability of our method to realistic cases.
>
> We include these experimental results in the “Deep Dive” section of the main text in our revision.
>
> ###  **Ablation Study on the RL Module and Baselines Handling Long-Term Memory:**
> We add an additional ablation study in the revision, in which we replace our RL module with static causal discovery baselines. The ablation results show that our RL module outperforms these baseline methods.
>
> However, because the long-term carry-over effect problem we consider—where historical values may have **direct** causal effects on the present—is relatively novel, we didn't find existing methods capable of handling such long-term **direct** causal effects. The baselines included in our work are able to handle long-term **indirect** causal effects.
>
> ### **Numerical Stability Under Near-Collinearity of Column Vectors:**
> In our system model, we assume that $X[t]$ contains independent noise $\epsilon(t)$, which makes the column vectors of $X$ highly unlikely to be nearly collinear. Therefore, we did not apply normalization or regularization during the QR decomposition step.
>
> ### **Redrawing Figure 3:**
> Thank you for the feedback. We redraw Figure 3 in the revision to enhance its clarity and interpretability.
>
> ### **References:**
>
> [1] Sergey Plis et al. Rate-agnostic (causal) structure learning.
>
> [2] M. Abavisani et al. GRACE-c: Generalized rate agnostic causal estimation via constraints.
>
> [3] Liu et al. Causal discovery from subsampled time series with proxy variables.
>
> [4] A Bellot et al. Neural graphical modeling in continuous-time: consistency guarantees and algorithms.

---

### Author Response · Authors · 2025-12-04
**Final Remark**

We sincerely thank the PC, AC and the reviewers for their thoughtful engagement and valuable feedback.

In the earlier discussion, we carefully addressed every comment raised by the reviewers. We clarified their concerns based on the content of our paper, added several new experiments to provide stronger evidence for our claims, and enriched the manuscript according to the reviewers’ suggestions. Unfortunately, we were unable to engage in further discussion with the reviewers. Below, we summarize what the reviewers appreciated about our work, how we addressed their main concerns, and the major changes made during the revision. We hope this summary will help the PC and AC better understand our work and the revisions we have undertaken.

**What reviewers appreciated about our work**:
1. Effectively addresses a critical and underexplored problem in temporal causal discovery: long-term historical confounding, with a feasible mechanism to eliminate historical effects. (Reviewer EqNz)
2. Introduces a novel QR-based score function backed by clear, provable theoretical results and strong interpretability. (Reviewer eY1p, 1PFq, Kj1Y)
3. Delivers clear motivation and presentation, innovative engineering implementation, rich simulation experiments, strong empirical performance, and high reproducibility. (Reviewer eY1p, Kj1Y, EqNz)

**How we addressed reviewers' main concerns**:
1. **Clarification on the Scope of Our Work**. Reviewer 1PFq raised the concern that our abstract and introduction give the impression of solving a full causal discovery problem, whereas later sections appear to focus only on causal skeleton recovery. **We respectfully clarify that our proposed method addresses the complete causal discovery problem, including the identification of causal directions.** This objective and graphical representation are consistent with mainstream approaches in temporal causal discovery.
2. **Theoretical Coherence between Static and Temporal Settings**. We appreciate Reviewer 1PFq’s scrutiny regarding the connection between the static analysis (Sections 3.1–3.3) and the temporal setting (Section 3.4).  **We would like to emphasize that the goal of our work is to recover the summary graph from time-series data, and the static analysis serves as the theoretical foundation rather than being a disjoint component.** To be specific, we first propose a scoring function and establish causal identifiability under this score in the static setting (Sections 3.2 and 3.3). In Section 3.4, we further show that the confounding effect of long-term history can be mitigated through a historical refinement scheme, which transforms the temporal causal discovery problem into static one. Consequently, we develop the LEVER algorithm, which applies the results from Sections 3.2 and 3.3 to the refined time-series data to address the temporal causal discovery problem.
3. **Distinctiveness of Our Contribution**. In response to Reviewer Kj1Y's concern that our setting substantially overlaps with existing causal structure learning work, **we underline that our primary contribution is distinct: we tackle causal discovery in temporal settings with long-term carryover effects.**  The discussion of the static scenario is not a repetition of existing work but a necessary step to establish the identifiability of our scoring function, which is then applied for causal discovery in temporal scenarios.
4. **Comparing with Existing Temporal Causal Discovery Methods**. In the related work section, we comprehensively review both classic and state-of-the-art temporal causal discovery methods. We emphasize that existing approaches typically assume a finite maximum time lag, whereas our work permits historical values at arbitrarily distant (even infinite) lags to directly influence current variables. This long-term carry-over effect introduces severe confounding that standard methods cannot handle. Following Reviewer 1PFq’s suggestion, we have added discussion of additional related studies. Following Reviewer eY1p’s advice, we also explicitly compare our work with methods designed for subsampling or continuous-time settings. We clarify the fundamental differences in goals and assumptions: in those works, the influence of distant history is mediated by observed intermediate variables and thus does not confound discovery; in our setting, distant history directly affects current values, requiring specialized treatment to remove historical confounding.
5. **Comparing with Existing Static Causal Discovery Methods**. Our causal discovery objective in the static setting is aligned with that of mainstream static causal discovery methods, yet our approach achieves better empirical performance. One likely reason is that our method is less sensitive to sample size than existing baselines. To substantiate this, we conducted new experiments in the static setting using 100, 200, 300, and 400 samples, confirming the superior robustness of our approach.
(To be continue)

---

> ### Author Response · Authors · 2025-12-04
>
> **Summary of Modifications in the Revision:**
> 1. Following Reviewers eY1p and 1PFq’s suggestions, we have substantially expanded the discussion of existing temporal causal discovery methods and subsampling-related works, with a deeper analysis of how our approach fundamentally differs from them.
> 2. We have incorporated two new studies into the Deep Dive section: (i) an ablation study on RL module, and (ii) a sensitivity analysis to violations of our key assumptions (previously located in the appendix). Additionally, the original window-size impact experiments have been moved from the Deep Dive section to the appendix, while we retain the heatmaps of $g(\tau)$ for different time lags as an informative visualization of long-term causal dependencies. These heatmaps demonstrate that the length of historical data required by our method can be considerably shorter than the duration over which carry-over effects actually persist. We have also included new experiments in the appendix on the static setting with varying sample sizes (100, 200, 300, and 400) to explain why our method outperforms existing static causal discovery baselines.
> 3. We redraw Figure 3 in the revision to enhance its clarity and interpretability.
> 4. We have refined the expression of our goal.

---

### Meta-Review · Area_Chair_aCd4 · 2026-01-05

**Summary:**

This paper proposes LEVER, a method for causal structure recovery in time series with long-term carry-over effects, combining QR-based order scoring with an efficient RL search. The work presents clean identifiability results, a principled limited-history refinement to mitigate historical confounding, and a compact RL state representation that yields substantial computational benefits. Reviewers find the theory sound and the empirical results strong, with consistent gains over static and temporal baselines on synthetic data and a real-world dataset. The approach is well motivated, technically solid, and appears reproducible.

Some limitations remain.
- The contribution is more clearly centered on static ordering with temporal refinement than on fully general temporal causal modeling, and the paper should better position itself relative to recent rate-agnostic and continuous-time causal discovery methods.
- The linear-recurrence assumption is strong and would benefit from robustness analysis, and additional ablations and temporal baselines could further strengthen the evaluation.
- The time-invariant skeleton assumption and the Stationary lag dependency should be clarified in detail, rather than following up existing literature without carry-over effects.

Overall, this is a solid and well-executed contribution with clear theoretical and practical value. I recommend acceptance as a poster.

**Reviewer Concerns:**

Nearly most reviewers' concerns should be addressed after the rebuttal.

**Reviewer Scores:**

I think that Reviewer Kj1Y (initial rating 4) will increase his score after the rebuttal.

---

### Decision · Program_Chairs · 2026-01-26

Accept (Poster)